# Shock wave impact on the viability of MDA-MB-231 cells

Yingqian Liao[1], James W. Gose[1], Ellen M. Arruda[2,3,4], Allen P. Liu[2,3,5,6], Sofia D. Merajver[7], Yin Lu Young[1,2,8]*

1 Department of Naval Architecture and Marine Engineering, University of Michigan, Ann Arbor, Michigan, United States of America, 2 Department of Mechanical Engineering, University of Michigan, Ann Arbor, Michigan, United States of America, 3 Department of Biomedical Engineering, University of Michigan, Ann Arbor, Michigan, United States of America, 4 Program in Macromolecular Science and Engineering, University of Michigan, Ann Arbor, Michigan, United States of America, 5 Cellular and Molecular Biology Program, University of Michigan, Ann Arbor, Michigan, United States of America, 6 Department of Biophysics, University of Michigan, Ann Arbor, Michigan, United States of America, 7 Division of Hematology and Oncology, Department of Internal Medicine, University of Michigan, Ann Arbor, Michigan, United States of America, 8 Department of Aerospace Engineering, University of Michigan, Ann Arbor, Michigan, United States of America

* ylyoung@umich.edu

**Data Availability Statement:** All relevant data are within the manuscript and its Supporting Information files.

**Funding:** This work was supported by MCubed project (Project ID: 826), University of Michigan.

## Abstract

Shock waves are gaining interests in biological and medical applications. In this work, we investigated the mechanical characteristics of shock waves that affect cell viability. *In vitro* testing was conducted using the metastatic breast epithelial cell line MDA-MB-231. Shock waves were generated using a high-power pulse laser. Two different coating materials and different laser energy levels were used to vary the peak pressure, decay time, and the strength of subsequent peaks of the shock waves. Within the testing capability of the current study, it is shown that shock waves with a higher impulse led to lower cell viability, a higher detached cell ratio, and a higher cell death ratio, while shock waves with the same peak pressure could lead to different levels of cell damage. The results also showed that the detached cells had a higher cell death ratio compared to the attached cells. Moreover, a critical shock impulse of 5 Pa·s was found to cause the cell death ratio of the detached cells to exceed 50%. This work has demonstrated that, within the testing range shown here, the impulse, rather than the peak pressure, is the governing shock wave parameter for the damage of MDA-MB-231 breast cancer cells. The result suggests that a lower-pressure shock wave with a longer duration, or multiple sequential low amplitude shock waves can be applied over a duration shorter than the fundamental response period of the cells to achieve the same impact as shock waves with a high peak pressure but a short duration. The finding that cell viability is better correlated with shock impulse rather than peak pressure has potential significant implications on how shock waves should be tailored for cancer treatments, enhanced drug delivery, and diagnostic techniques to maximize efficacy while minimizing potential side effects.

The grant was received by Prof. Yin Lu Young, Prof. Ellen Arruda, and Prof. Sofia Merajver. The funders had no role in study design, data collection and analysis, decision to publish, or preparation of the manuscript.

**Competing interests:** The authors have declared that no competing interests exist.

## Introduction

Shock waves are supersonic pressure waves with a high amplitude and a short pulse duration. Shock waves can be generated through a shock tube, an extracorporeal shock wave lithotripsy, or a laser. Shock waves have been applied to many fields in medical applications, including drug delivery [1–3], gene transfer [4], treatment of stone diseases, and bone and tendon disorder therapies [5–7]. Shock wave technique has the potential advantages of being a non-invasive, targeted, extracorporeal cancer treatment method [8]. Hence, it is important to study the effect of shock wave on biological tissues and cells.

Previous studies have exploited the interaction between shock waves and biological tissues and cells. Shock waves have been shown to change cell membrane permeabilization through the shear force induced by the relative motion between a target and surrounding fluid, and thus induce uptake of molecules and drugs [4,9]. The change in membrane permeabilization introduced a new means to overcome the blood-brain barrier (BBB) to deliver a drug to the targeted brain region [3]. In stone disease treatment, tensile stress exerted by the shock waves can lead to cavitation, and the bubble dynamics causes fluid jets. The shock waves and the induced cavitation dynamics lead to significant local sound field change and energy exchange, which could be strong enough to break calcified tissues [5,6]. It was also observed that shock waves induce tissue and cell damage [10–12]. Shock waves with peak pressures as low as 1 MPa were found to cause mild cell morphology changes in a rat's brain [13]. Gamarra *et al.* found that high-energy shock waves can achieve similar level of tumor remission compared to surgery using *in vivo* experiments [14]. Several other works also show that shock waves induce damage on tumors [15,16].

In previous work, the involvement of heat and cavitation dynamics complicated the study of shock waves' impact on the cell viability, as they can couple with mechanical stresses to cause cell damage and are difficult to control. However, some experiments [10,17,18] suggested that biological effects happened even without the occurrence of cavitation, and pointed to the importance of other mechanical effects of shock waves, including the peak pressure, the rise time, and the shock wave impulse. Schmidt *et al.* conducted *in vitro* experiments to investigate shock wave effects on U87 brain cancer cells. They found that when the incident pressure exceeds a lethal level, shock waves can cause significant cell damage [19]. Most of the previous work focused on correlating cell damage to the peak pressure and the stress gradient [11,19], while few unveil directly the relation between shock wave impulse and cell damage. Impulse is the integral of pressure over time, as shown in the following equation,

$$I = \int_0^\infty P dt \qquad (1)$$

The impulse represents the kinetic energy transferred from the shock waves to the target. As shown in the analytical study presented in Liu and Young [20] and Li *et al.* [21], the momentum of rigid and deformable solids when reaching the peak response is governed by the impulse when the shock event is extremely short. Kodama *et al.* [22] studied the effect of shock wave impulses on cell permeabilization using human promyelocytic leukemia cells. They found that controlling the impulse at a certain range could increase the molecular uptake without causing cytotoxicity, while the shock waves with higher peak pressures but much smaller impulses resulted in negligible fluorescence uptake. A previous work [23] on design of armor has shown that impulse could play a more important role than peak pressure in causing damage to target organisms if the duration of the pressure wave ($T$) is shorter than the fundamental natural period of the target, $T_{target}$, i.e. $T < T_{target}$. This was demonstrated by using a simplified dynamical system with two degrees of freedom, and it was shown that the maximum pressure on the target depends on the transmitted impulse when the transmitted impulse is

shorter than $T_{target}$ [23]. Shock waves with the same impulse can be generated in different ways, as shown in Fig 1. This suggests the potential to design shock waves with desirable characteristics to increase or mitigate biological damage, depending on the need.

The objective herein was to investigate the effect of different mechanical characteristics of shock waves on cell viability and identify the governing mechanical parameter or parameters, which is helpful in guiding the tuning of shock waves in medical applications, such as cancer treatment, kidney and gall stones treatments, enhanced drug delivery and diagnostic techniques [24,25]. Laser-induced shock waves were used to investigate the effect on cell viability by varying the shock wave pressure and impulse. Laser-induced shock waves can generate pressure waves with high repeatability, and minimize heat effects, as the duration is too short to cause significant heat accumulation if only a few pulses are generated. Moreover, laser-induced shock waves exclude the side effect of cavitation [19,26]. By utilizing this technique, we found that shock waves with higher impulses led to decreased cell viability, while shock waves with similar peak pressures could have different effects on cell viability.

## Materials and methods

### Laser-induced shock wave generation

The setup used to produce the shock waves was similar to that described by Schmidt *et al.* [19]. The method used here has been long utilized in metal processing and for the study of the dynamic properties of materials [27–29]. Fig 2(A) demonstrates the shock waves generation process. When a laser pulse strikes the absorption layer surface, the high-intensity and short duration laser beam heats a thin layer of the material and then induces thermal diffusion. The interaction time is so short that the thermal diffusion accumulates in a confined volume. When the vaporization of the absorption material reaches the critical energy, the absorption material turns into plasma because of the ionization. The plasma strength increases until the energy deposition completes. The plasma expansion by reaction generates pressure pulses for the shock initiation and the induced shock waves propagate in the direction opposite to that of plasma diffusion. The transparent substrate is used to confine the plasma expansion so that the shock waves are strengthened [30,31].

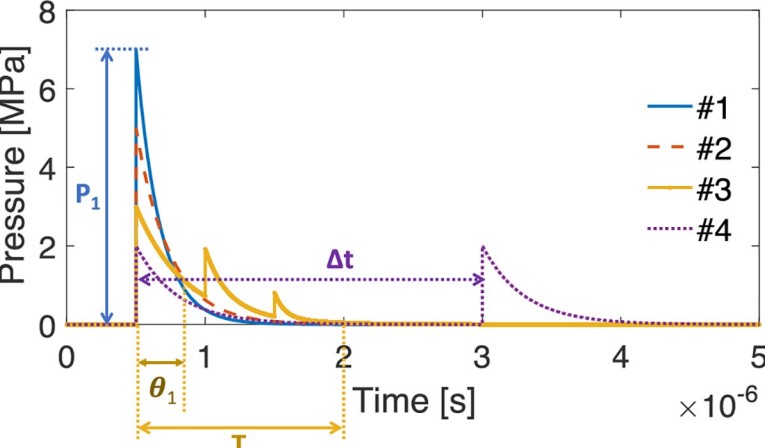

**Fig 1. Four different shock wave profiles that produce the same impulse.** $P_1$ is the peak pressure. $\theta_1$ is the decay time of the first peak. T is the total duration of the shock waves. $\Delta t$ is the time gap between two sequential shock waves. The area under the shock wave curve represents the impulse.

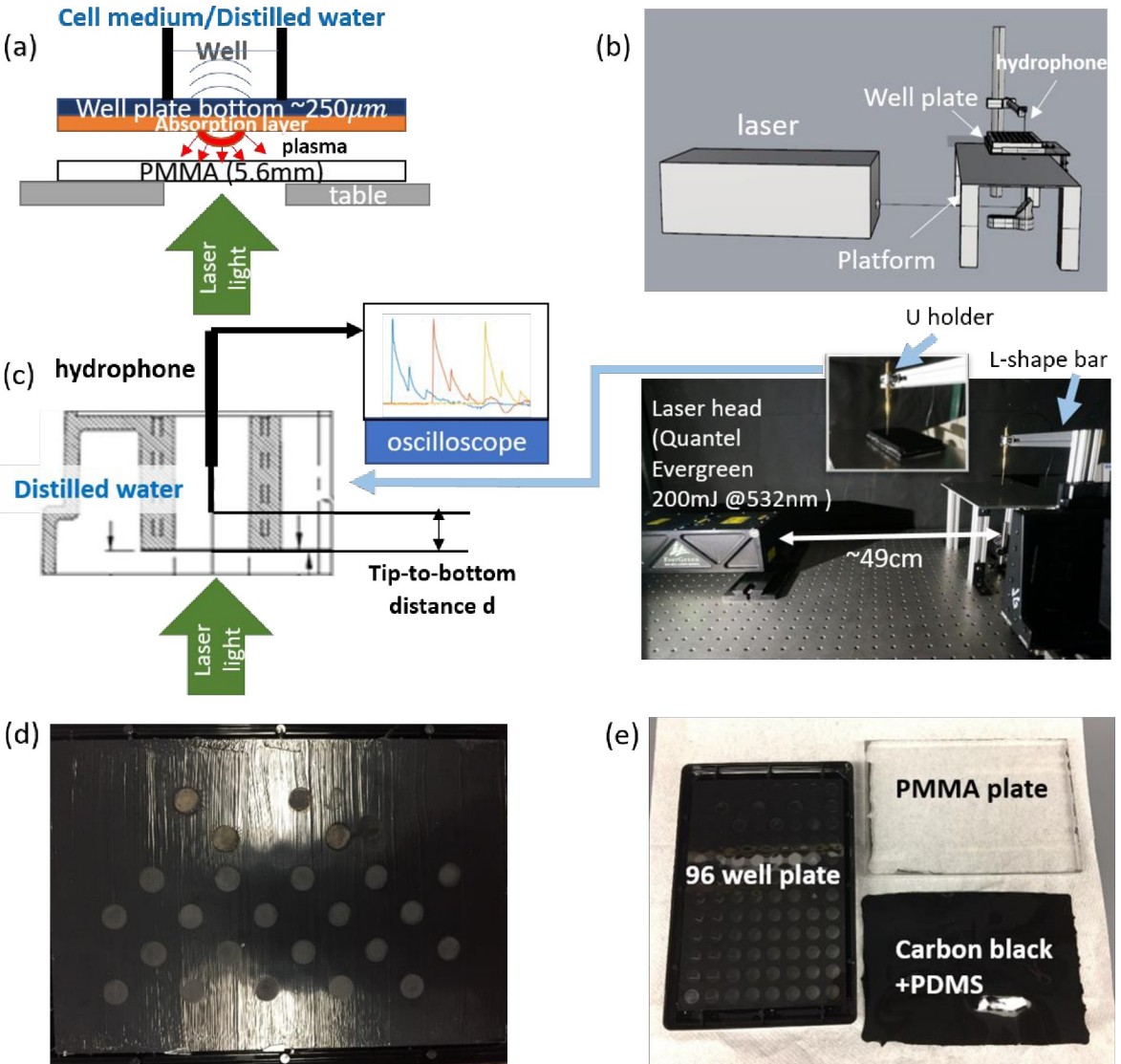

**Fig 2.** (a) Illustration of the laser-induced shock wave mechanism on one of the wells. (b) 3-D model of the experimental setup. (c) Left: Details on the pressure measurement using the hydrophone. The tip-to-bottom distance refers to the distance between the tip of the hydrophone and the top surface of the well bottom. Right: The overall layout of the experimental setup. (d) and (e) Plate treatment method illustrations. (d) The plate bottom coated with art paint with polydimethylsiloxane (P/PDMS). The grey circles are areas illuminated by laser. (e) The plate bottom coated with carbon black mixed polydimethylsiloxane (C/PDMS).

## Experimental setup

Fig 2(B) and 2(C) show the overall experimental layout. A Quantel Evergreen 200 (Quantel, USA) was used as the laser beam source. The laser had pulse energy up to 200 mJ and a repetition rate up to 15 Hz with a 532 nm wavelength. A Nunc™ MicroWell™ 96-Well Optical-Bottom Plate (Nunc™ MicroWell™, 165305) with Polymer Base was used as the cell culture dishes. The well plate is shown in Fig 2(C) and 2(E). Cells were pre-cultured in the well plate before conducting the experiment. The well plate was placed on a small aluminum platform fixed on the optical table. A 6 mm diameter hole was drilled in the middle of the aluminum platform to allow for the laser light passage. The hole diameter was determined by the diameter of the laser beam light in the near field. An optical mirror was placed at an angle of 45° under the

aluminum platform to re-direct the laser beam vertically. The aluminum L-shape bar shown in Fig 2(C) was used to hold the hydrophone for pressure measurements during the experiment. The L-shape aluminum bar was attached to assembled translation stages with two degrees of freedom–vertically and laser-beam wise. Another degree of freedom was provided by moving the U holder along the slide in the aluminum bar.

## Well plate pretreatment methods

In the present work, two different approaches were used to pre-treat the well plate to generate shock waves with different profiles. One approach utilized art paint with polydimethylsiloxane (P/PDMS) coating and the other utilized carbon black mixed polydimethylsiloxane (C/PDMS) coating. The treated plates are shown in Fig 2(D) and 2(E). The paint used in P/PDMS coating was Martha Stewart Crafts Multi-Surface High Gloss Acrylic Craft Paint in Assorted Colors (2-Ounce) with black color. A dry paintbrush was used to apply paint on the bottom surface of well plates. First, a single layer of paint was applied and then dried for 5 minutes. Next, it was followed by a second layer of paint thicker than the first. Care was taken to ensure the paint was even and thick enough such that light could not be seen through. Each well was treated with a single pulse only. Hence, as long as the paint layer was thick enough such that a single laser pulse did not over-burn the paint, the resultant shocks between wells and plates should be comparable, which is shown later in Fig 3(A). The paint was allowed to cure for two days. After the curing, the polymethylmethacrylate (PMMA) plate was attached to the well plate bottom using PDMS. The PDMS was chosen as it is bio-friendly and easy to degas. 0.5 g of curing agent from the Sylgard 184 kit was added to a 20 ml disposable vial, followed by 5 g of PDMS. The mixture was mixed thoroughly with a 1 ml pipette tip. A vacuum chamber was necessary to degas and remove the bubbles from the treated material, which created a clear path for the laser beam. The degassed PDMS was poured carefully onto the well plate bottom. The well plates with PDMS were put back into a vacuum chamber again for the second degassing. The PMMA plates were carefully placed on the PDMS following the removal of the air bubbles. The PMMA plates were placed into the oven to heat at 70˚C for 2–3 hours after the PDMS had spread across the entire contact area. The oven was turned off after 2–3 hours and the well plates were kept in the oven until the oven temperature reached room temperature. The second approach utilized a C/PDMS coating. To produce the C/PDMS coating, 0.275 g of carbon black was added into a 20 ml disposable vial. The carbon black particles were ground with a lab spoon into a powder form. 0.5 g of curing agent from the Sylgard 184 kit was then added to the 20 ml disposable vial, followed by adding 5 g of PDMS. The mixture was stirred with a 1 ml pipette tip. A sonicator was used to mix the carbon black and PDMS. After mixing the carbon black with PDMS, the same procedure as the P/PDMS coating but without the painting was followed.

## Shock wave pressure measurement

The shock wave pressure time histories were recorded using a needle hydrophone, an attenuator, a preamplifier, and a DC coupler (Precision Acoustics Ltd., UK). The measurement was processed in distilled water. The hydrophone was capable of measuring frequencies ranging from 10 kHz to 60 MHz. The hydrophone was calibrated from 1 MHz to 30 MHz. The sensitivity at 30MHz was used to calculate the pressure, which was 26 mV/MPa. The sensitivity has an uncertainty of 17% according to the calibration data from the manufacturer; however, smaller uncertainties were observed at greater pressures. The tip-to-bottom distance in Fig 2 (C) refers to the distance between the hydrophone tip and the top surface of the well bottom. Pressure measurements were conducted with a tip-to-bottom distance of d = 2.1 mm except

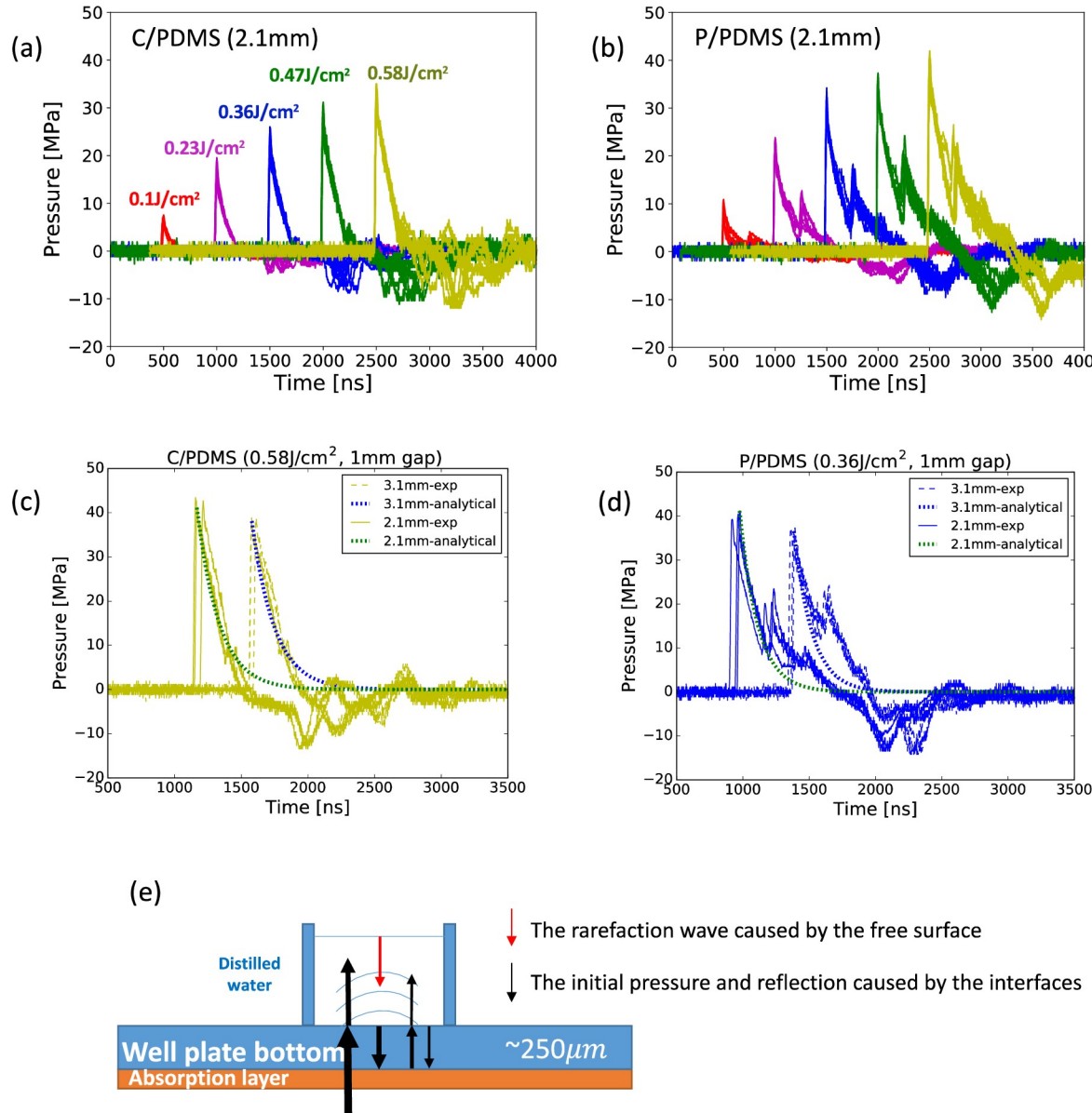

**Fig 3.** (a) Recorded pressure time histories for the C/PDMS coating. (b) Recorded pressure time histories for the P/PDMS coating. In (a) and (b), the time histories in each plot show nine measurements from three independent tests. Each time history of the same laser fluence was offset to align the first peak. Note that in-house calibration data was used. The sensitivity used here is 26mV/MPa and has an uncertainty of 17%. For both the P/PDMS and C/PDMS coatings, the measurements were taken at laser fluences of 0.11, 0.24, 0.36, 0.48, and 0.58 J/cm2 respectively. The curves with the same color are results from the same laser fluence. (c) and (d) show the raw data of pressure time histories measured at two different heights with a difference of 1mm, which was used to estimate the velocity of the shock waves. The results for C/PDMS coating were given at pulse energy 0.16J, while the results for P/PDMS coating were given at pulse energy 0.1J. Analytical exponential decay functions using the first peak pressure and the decay time of the first peak were shown as a reference. (e) Schematic illustration of the cause for the multiple shocks and the significant negative pressure.

where noted. Each well was used only once for a single pressure measurement. The laser was triggered by a waveform generator, and the hydrophone signal was recorded with a Tektronix DPO3014 oscilloscope (Tektronix, Inc.). To measure the pressure at a specific tip-to-bottom distance, the well plate position was first adjusted so that the hydrophone tip was placed at the middle of the well. Aluminum foil with precision ground to 0.3 mm was used to locate the

hydrophone tip relative to the top plane of the well. After lowering the vertical position of the hydrophone tip to be the same as the top of the well, the hydrophone tip was moved down by a specific distance to record the shock wave pressure time histories.

## Cell culture

We used metastatic breast epithelial cell line MDA-MB-231. Our stock of MDA-MB-231 cells were obtained from Dr. Janet Price at the MD Anderson Cancer Center. The original line has been authenticated by single tandem repeats (STR) analysis. Cells were cultured at 37°C with 5% $CO_2$ in 10 cm dishes in RPMI 1640 medium supplemented with 10% fetal bovine serum, 1% penicillin-streptomycin and Gentamicin. For the WST test (WST reagent from Sigma-Aldrich), cells were cultured in pretreated 96 well plates at an initial density of $2\times10^4$ cells/well with 100 μL medium in each well, and then incubated at 37°C, 5% $CO_2$ condition for 30–40 hours. Subsequently, five laser fluences were tested for both the P/PDMS and the C/PDMS coatings. The laser impulse was applied to four different wells for each laser fluence.

For the imaging test, cells were cultured in the pretreated well plates at an initial density of $1\times10^4$ cells/well with 200 μL medium in each well to yield a good imaging density after laser treatment, and then incubated at 37°C, 5% $CO_2$ condition for 36–38 hours. Fresh media was added before laser treatment since the cells were cultured for one more day after laser treatment in order to assess variation of cell viability with time. Three different laser fluences were tested for both the P/PDMS and the C/PDMS coatings. For each plate, eight wells were treated at the same laser fluence. Among the eight wells, four samples were stained with propidium iodide and Hoechst right after laser treatment to investigate the immediate cell damage, while the remaining four samples were stained 24 hours after laser treatment for the investigation of the delayed damage.

## Laser treatment

The hydrophone tip was calibrated to be at the center of the hole on the aluminum platform, which was additionally used to align and evenly distribute the laser beam through a specified area. The well plate was placed on the aluminum table and was adjusted to have the hydrophone tip pointing at the center of the test well. A waveform generator was used to trigger the laser to give a single pulse for each treatment. A sealing film was used to cover the top openings of the well plate to minimize medium lost from the well due to shock wave effects. An air incubator was used to keep the temperature of the wells around 37°C during laser treatment.

## WST cell viability test

The WST-1 colorimetric assay was used to quantify cell proliferation, cell viability, and cytotoxicity. The working volume of the WST-1 was 10% of the total medium. After adding the WST-1 reagent, the cells were incubated at 37°C for one hour. The absorbance was read using Cytation 5 (Biotek Instruments) with 450 nm excitation. WST absorbance results were shown as normalized values, where data was normalized by the absorbance value of the control group, i.e. the group without the shock treatment.

## Propidium iodide and Hoechst stain

Propidium iodide (PI, Sigma-Aldrich Inc., MO, USA) and Hoechst (Hoechst 33342, Invitrogen) stain were used for labeling nuclei of dead cells and all cells, respectively. A stock concentration of 1.5 mM PI was diluted to a working concentration of 42 μM. A 30 mg/mL Hoechst stain reagent was diluted to a concentration of 1 mg/mL, and a final concentration of 1 μg/mL

was used. Both diluted solutions were stored at -20˚C. After adding the dye reagent, cells were incubated for 40 mins at 37˚C and 5% $CO_2$.

## Imaging and image analysis

Fluorescence imaging was performed by using Cytation 5 Plate Reader. Three different channels were used: phase contrast, red fluorescence protein (RFP), and DAPI. The optimal excitation and emission wavelengths for PI are 535 nm and 617 nm, respectively. The optimal excitation and emission wavelengths for Hoechst stain are 361 nm and 486 nm, respectively. Four ($2 \times 2$ montage) 4x images were taken using auto reading mode for each well. A relatively low magnification was used for imaging to capture a larger cell coverage area, as the intent of the current analysis is on the influence of shock wave impact on cell viability, and not to investigate the effect on detailed cell morphology.

Image analyses were performed using Gen5 (Biotek Instruments, Inc., VT, USA), which was integrated with the Cytation 5 multi-mode plate reader. The cellular analysis in the statistical analysis was used for cell counting. The cell count from the DAPI channel represented the number of total cells and the cell count from the RFP channel represented the number of dead cells. The minimum size and the maximum size used for cell counting were 10 μm and 50 μm respectively. The images were stitched together for the cell count for each well. The stitched image covered 21% of the well bottom area. Since auto imaging was used, the imaging locations were the same for all wells.

The detached cell ratio was calculated by dividing the number of detached cells by the sum of detached cells and attached cells. The cell death ratio was calculated by dividing the number of dead cells by the number of total cells. The number of live cells was computed by the number of cells counted from the DAPI channel after subtracting the number of cells counted from the RFP channel.

## Results

### Pressure profile comparison between two different treatments

Fig 3(A) and 3(B) show the pressure time histories for both the P/PDMS and C/PDMS coatings. The measurements in Fig 3(A) and 3(B) are from three sets of independent tests with three samples in each independent test, one sample per well. Good repeatability is observed between the different independent tests, which suggests minimal impact from the potential uneven well coating treatment.

As shown in Fig 3(A) and 3(B), the profiles of a single shock pressure from both treatment methods are of the same shape, which can be described analytically using an exponential decay function. The analytical exponential decay function $P = P_1 e^{-\frac{t-t_0}{\theta}}$ using the first peak pressure $P_1$ and the decay time $\theta$ of the first peak are plotted with the measured time histories in Fig 3 (C) and 3(D). The analytical exponential decay overlaps with the first peak well, which indicates the analytical expression can be used to describe the shock waves.

To estimate the velocities of the shock waves, we measured the pressure at two different tip-to-bottom distances, 2.1 mm and 3.1 mm, as shown in Fig 3(C) and 3(D). The time gap between the two first pressure peaks measured at these two distances represents the time that the shock waves took to travel 1 mm. The approximate time interval for the C/PDMS coating was 407±4 ns, while that for the P/PDMS coating was 433±4 ns. Accordingly, the speed of shock waves for the C/PDMS coating with pulse energy of 0.16 J was 2457±125 m/s. The speed of shock waves for the P/PDMS coating with pulse energy of 0.1 J was 2304±117 m/s. For both

cases, the pressure waves travelled faster than the speed of sound in water (1498 m/s), which confirms that they are supersonic shock waves.

Fig 3(A) and 3(B) show that, for both coatings, there exist multiple peaks in the time histories. The strengths of the subsequent peaks of the P/PDMS coating were higher than the C/PDMS coating. The subsequent peaks in each of the pressure time histories shown in Fig 3(A) and 3(B) were caused by reflection as a result of the impedance mismatch between water and the well plate bottom, and the impedance mismatch between the well plate bottom and the absorption layer, as shown in Fig 3(E). From Fig 3(B), the time gaps between the first peaks and the second peaks for all laser fluences were approximately constant at 252–300 ns.

For both coatings, negative pressures are observed in the tail end of the time histories in Fig 3(A) and 3(B), which could be caused by cavitation. Nevertheless, note that the amplitudes of the maximum negative pressures were much lower than the vapor pressure of water, suggesting that the negative pressure is probably caused by rarefaction waves reflecting back from the free surface, the interface between water and air.

The pressure measurement data from four sets of independent tested were summarized in Fig 4. In each independent test, three wells were measured for each laser fluence. The first peak pressure $P_1$, the decay time of the first peak $\theta_1$ and the shock wave impulse $I$ are summarized in Table 1. The decay time is defined as the time for the pressure to decrease from the peak to the amplitude of 0.368 times peak pressure. The shock wave impulse was calculated by integrating the pressure history from the first peak front to when the pressure value becomes negative, which is given as follow,

$$I = \int_0^{t^+} P dt \qquad (2)$$

where $t^+$ is the end time of the positive phase of the shock wave.

Fig 4 and Table 1 show that at the same laser fluence, the values of the first peak pressure, the decay time, and the impulse for the P/PDMS coating are higher than those of C/PDMS coating. This variation was caused by the difference in the absorption layer. For the C/PDMS, the absorption layer was the mixture of carbon black and PDMS, while for the P/PDMS, it was purely the paint. Due to the limited maximum pulse energy of the laser used here (0.2 J), we did not detect the saturation point for peak pressure, decay time and impulse. However, we can see from Fig 4 that, for both coatings, the peak pressure and the decay time have already started to approach the saturation value. This is because the laser density became so high that

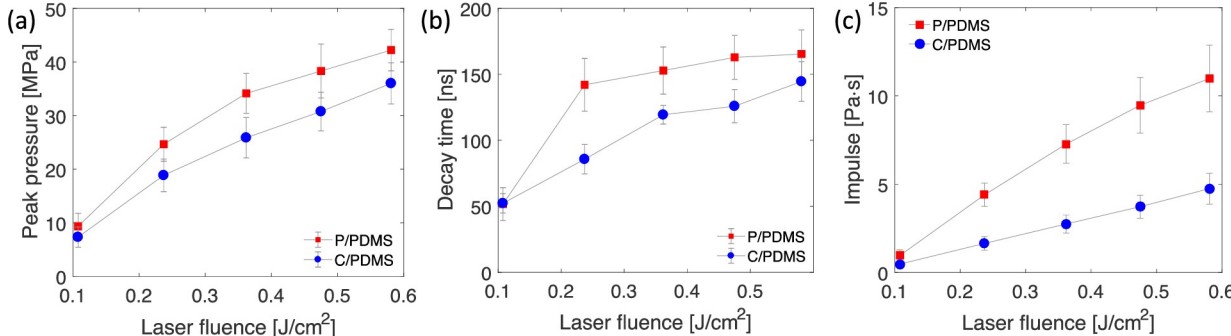

**Fig 4.** (a) Peak pressure versus laser fluence for the C/PDMS and P/PDMS coatings. (b) Decay time of the first peak versus laser fluence for the C/PDMS and P/PDMS coatings. (c) Impulse versus laser fluence for the C/PDMS and P/PDMS coatings. The plots in (a), (b) and (c) show results summarized from four independent tests. The error bars indicate the standard deviations from four independent tests with the same setup and conditions.

**Table 1. Summary of pressure measurements.**

| Pulse energy (J) | 0.03±0.00 | 0.07±0.00 | 0.10±0.00 | 0.13±0.00 | 0.16±0.00 |
|---|---|---|---|---|---|
| Laser fluence (J/cm$^2$) | 0.11±0.00 | 0.24±0.00 | 0.36±0.00 | 0.48±0.00 | 0.58±0.01 |
| P/PDMS $P_1$ (MPa) | 9.35±2.42 | 24.68±3.17 | 34.16±3.73 | 38.33±5.04 | 42.24±3.89 |
| C/PDMS $P_1$ (Mpa) | 7.33±1.90 | 18.87±3.04 | 25.88±3.76 | 30.78±3.60 | 36.03±3.84 |
| P/PDMS $\theta_1$ (ns) | 51.68±12.49 | 142.03±19.91 | 152.88±17.83 | 162.88±16.62 | 165.38±18.23 |
| C/PDMS $\theta_1$ (ns) | 52.33±7.43 | 85.69±11.17 | 119.38±7.05 | 125.88±12.60 | 144.58±15.08 |
| P/PDMS I (Pa·s) | 0.97±0.32 | 4.40±0.65 | 7.27±1.09 | 9.47±1.58 | 10.99±1.89 |
| C/PDMS I (Pa·s) | 0.48±0.11 | 1.65±0.38 | 2.75±0.51 | 3.73±0.65 | 4.75±0.87 |

Values reported are averages and standard deviations are reported here. At the same laser fluence, the resultant peak pressure, decay time, and impulse of P/PDMS coating are higher than those of C/PDMS. At similar peak pressure, such as P/PDMS with 0.36J/cm$^2$ (34.16 Pa) and C/PMDS with 0.58 J/cm$^2$ laser fluence (36.03 Pa), the impulse of P/PDMS (7.27 Pa·s) is higher than that of C/PDMS (4.75 Pa·s). Sample pressure time histories at different laser fluences can be found in Fig 2.

the breakdown of the confined materials occurred, so the laser energy was absorbed before it ablated the absorption layer. A similar observation was discussed by Fabbro *et al.* [28].

## Cell viability–WST absorbance results

In the WST test, the cells were analyzed as a population and the absorbance was read one hour after laser treatment by adding the WST-1 reagent. Since we are interested in how the shock waves affected cell viability, absorbance normalized by the control group value, i. e. the group without the shock treatment, was reported to show the changes in viability compared to the control group. At the lowest peak pressure, the absorbance of both coatings was the same as the control group, which is indicated by a normalized absorbance value around 1.0 as shown in Fig 5. As highlighted in the blue and red boxed values in Fig 5(A), at similar peak pressures around and over 30 MPa, higher normalized absorbance was observed for the C/PDMS coating, which indicates higher cell viability, compared to the P/PDMS coating. The difference in

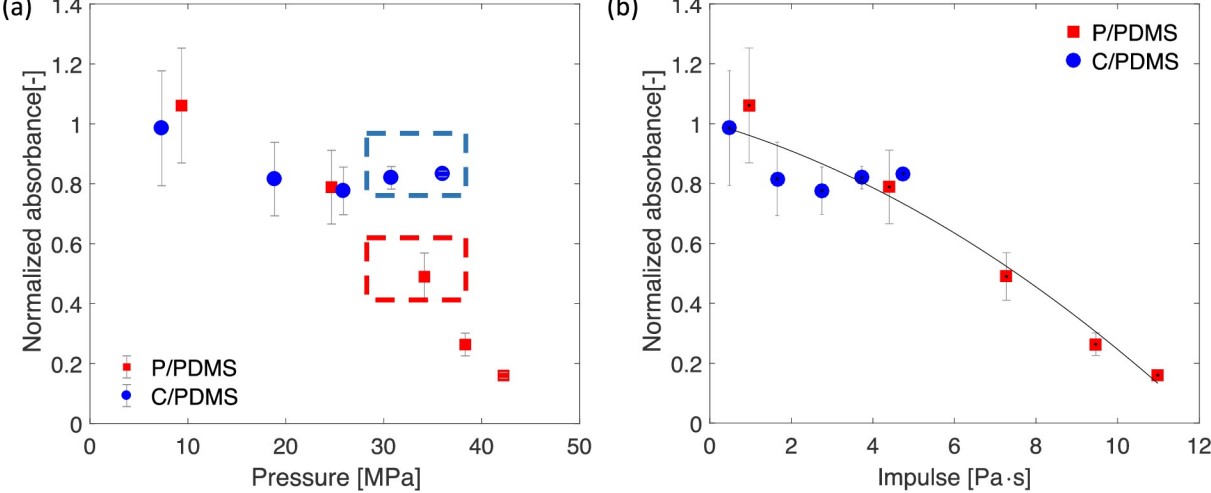

**Fig 5. WST test absorbance results for both P/PDMS and C/PDMS coatings at five different laser fluence values.** The absorbance values are normalized by the control group, which are untreated cells. (a) Normalized absorbance versus peak pressure. (b) Normalized absorbance versus impulse. The black line shows the fitted smoothing splines using all data points to show the overall trend. Results from three independent tests are shown. The error bars show the standard deviations from three independent tests, and each independent test include four wells for each laser fluence.

the normalized absorbance at similar peak pressures between the two coatings suggested that the peak pressure did not solely affect cell viability. On the contrary, when we used the impulse of shock waves to collapse the normalized absorbance results, as shown in Fig 5(B), the normalized absorbance for both coatings decreased monotonically with increasing impulse. When the impulse of the shock waves was below 5 Pa·s, the impact on the normalized absorbance is small, as shown in Fig 5(B). When the shock wave impulse value exceeded 5 Pa·s, a significant drop in normalized absorbance was observed. The difference in trends when the peak pressure exceeds 30 MPa shown in Fig 5(A) is caused by the difference in impulse values at similar peak pressures between the two coatings. For the P/PDMS coating, when the peak pressure reaches 30 MPa, the impulse reaches 5 Pa·s; while for the C/PDMS coating, the impulse is much lower with a similar peak pressure. Overall, the results in Fig 5 suggest that the impulse of shock waves is the parameter that best correlates with the viability of MDA-MB-231 cells.

Rahimzadeh *et al*. [23] have shown that when the duration of pressure waves is significantly shorter than the characteristic time of the dynamic response of the target, the impulse, instead of the pressure, is the parameter that governs the damage of the target. The dynamical characteristic time, or the fundamental natural period, of MDA-MB-231, can be estimated using Eq 3.

$$T_c = \frac{2\pi}{\sqrt{\frac{ER}{m}}}$$

(3)

where ~200 Pa is the Young's modulus of a MDA-MB-231 cell according to Lee and Liu [32]; $R = 10–15$ μm is the approximate radius of a single cell; $m = 10^{-12}$ kg is the approximate mass of a cell [33]. The estimated characteristic time of a MDA-MB-231 cell is $T_C = 115–140$ μs, which is more than two orders of magnitude higher than the maximum duration of the shock waves of $T = 1$ μs, as shown in Fig 3. Hence, the impulse, instead of the peak pressure, should be the critical parameter that governs the impact of the shock waves on cell viability since $T<<T_C$.

## Cell viability correlates with impulse instead of peak pressure of shock waves

In order to determine cell viability at the single cell level, we performed imaging by using PI and Hoechst stain. Fig 6(A) shows the workflow for the imaging tests. Since cells could detach after laser treatment, we separated the detached cells and attached cells to investigate the shock wave impact on cell detachment and on cell death. We changed the old medium before laser treatment to remove the initial dead cells and to keep the medium fresh for later incubation to investigate the cell death on the second day (day 1). The results of Hoechst stain from the imaging right after laser treatment (day 0) are shown here to investigate the shock impact on the percentage of cell detachment.

Fig 6(B) shows the detached ratio data right after laser treatment to compare the effect of peak pressure and impulse on cell detached ratio, which is the ratio of the number of detached cells to the sum of detached cells and attached cells. Fig 6(C) shows sample images on the detached cells. Fig 6(B) shows that for shock waves with similar peak pressures, the blue dashed box for C/PDMS and the red dashed box for P/PDMS, led to different detached cell ratios. Instead of the peak pressure, the impulse of the shock waves correlated best with the detached cell ratio. A higher detached ratio is observed with a higher shock wave impulse value. Some cells were forced to detach by the incident high pressure, while the cells that remained attached were also affected gradually by subsequent pressure, which is the

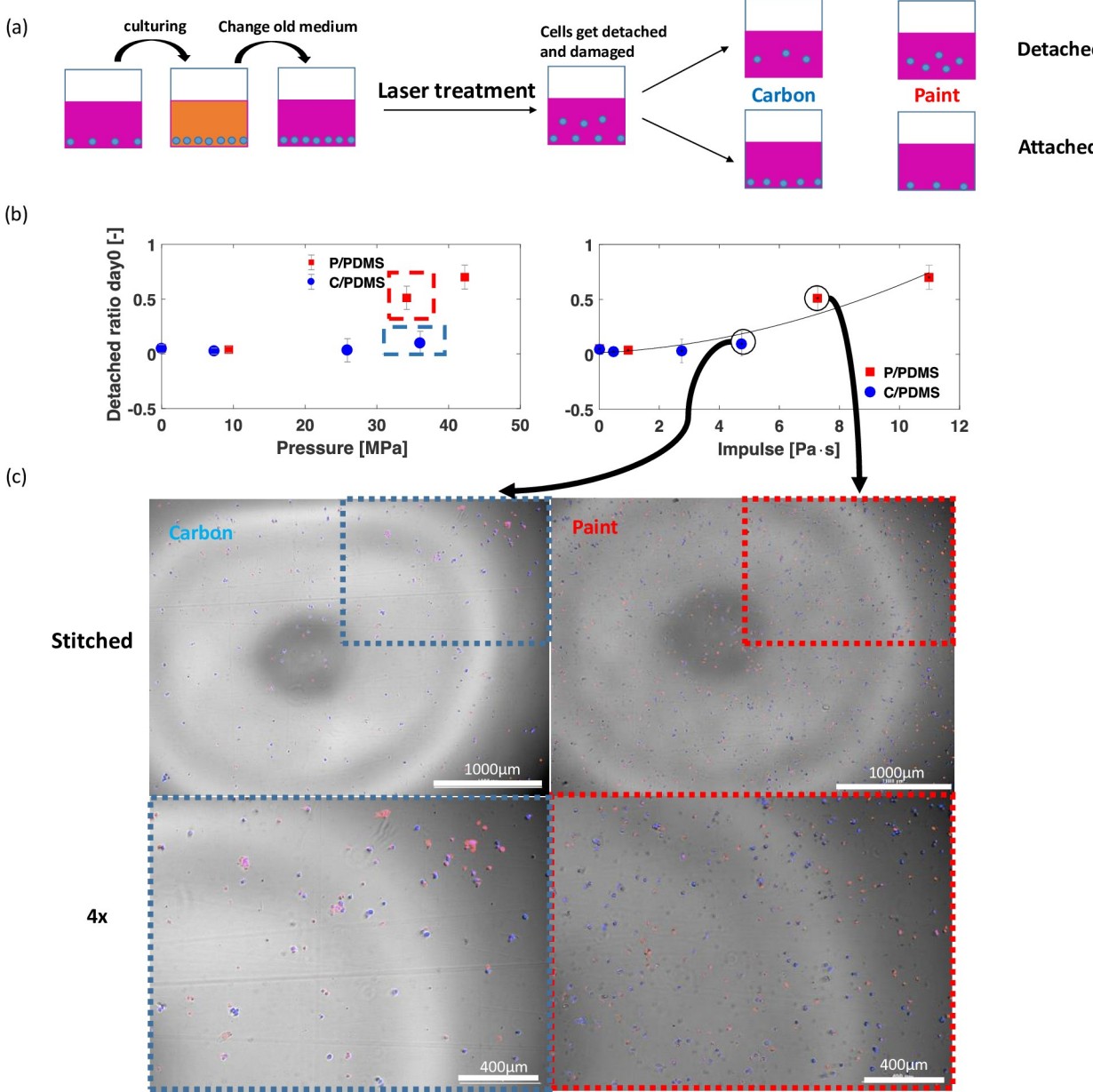

**Fig 6.** (a) Workflow for the imaging tests. The pink color represents the fresh medium while the maize color represents the old medium. The blue dots represent cells. (b) Detached cell ratio versus pressure and detached cell ratio versus impulse from imaging results for both P/PDMS and C/PDMS coatings. The error bars show the standard deviations. Results are from three independent tests and four samples were tested in each independent test. (c) Sample images from the circled cases are shown. The blue dots in the image represent the total detached cells. The red dots in merged images represent the dead cells.

accumulative effect represented by the shock wave impulse. When the shock wave impulse exceeded 5 Pa·s, the ratio of the detached cells increased significantly with a higher impulse, while shock waves with an impulse lower than 5 Pa·s showed a negligible influence on cell detachment. This suggests there exists a critical impulse value that will lead to significant cell damage.

We performed Hoechst and PI stain and imaging right after laser shock treatment (day 0) to investigate the immediate cell damage, and performed imaging one day after laser shock

treatment (day 1) to investigate the delayed cell damage. Fig 7(A)–7(D) shows the results of cell death ratio against the peak pressure, and Fig 7(E)–7(H) shows the cell death ratio against the impulse.

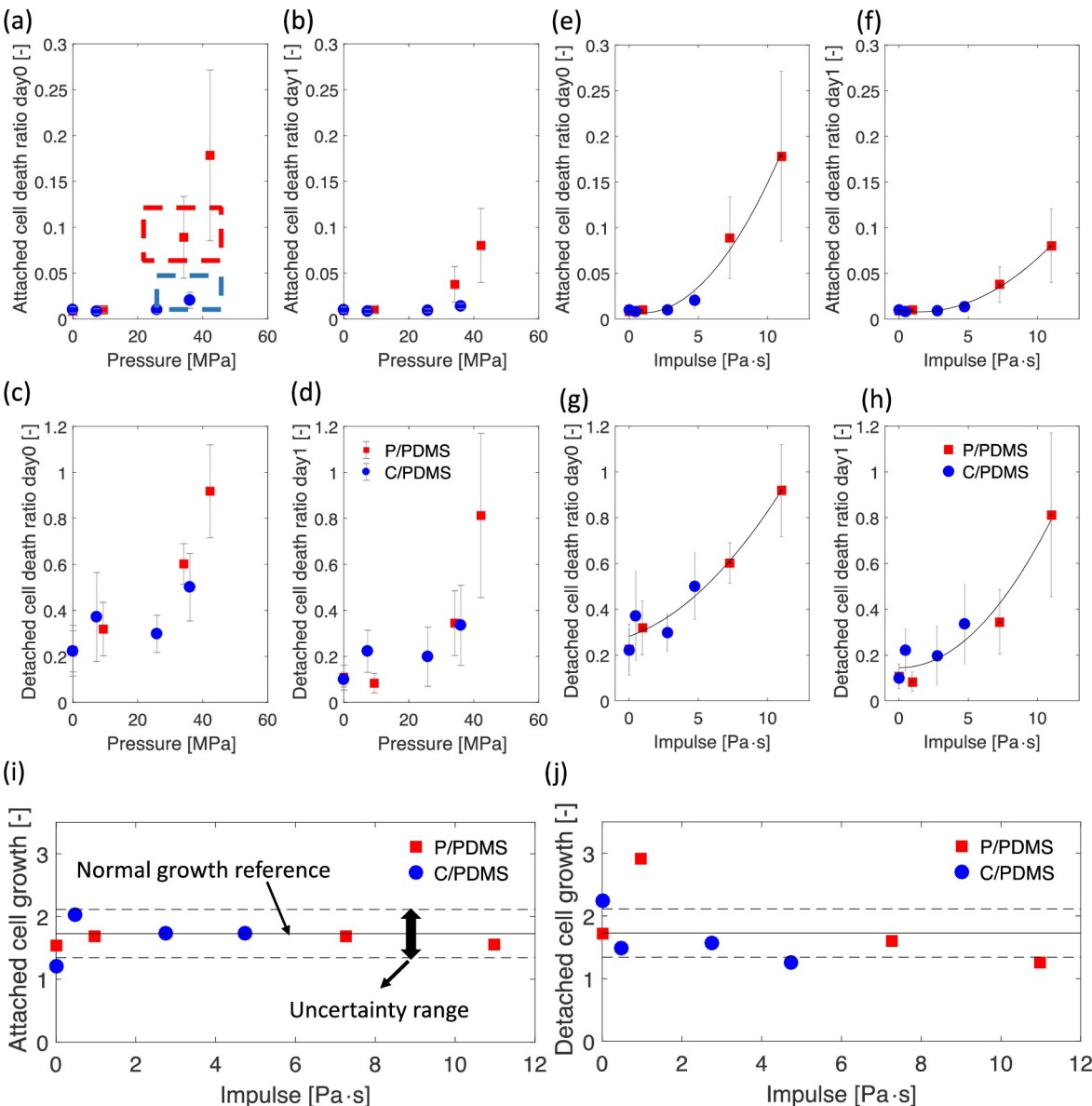

**Fig 7.** (a), (b), (c), and (d) Cell death ratio versus the pressure from imaging results for both P/PDMS and C/PDMS coatings. (a) Cell death ratio among attached cells on day 0. (b) Cell death ratio among attached cell on day 1. (c) Cell death ratio among detached cell on day 0. (d) Cell death ratio among detached cells on day 1. (e), (f), (g), and (h) Cell death ratio versus the impulse from imaging results for both P/PDMS and C/PDMS coatings. (e) Cell death ratio among attached cells on day 0. (f) Cell death ratio among attached cell on day 1. (g) Cell death ratio among detached cell on day 0. (h) Cell death ratio among detached cells on day 1. The error bars show the standard deviations. Results are from three independent tests and four samples were test in each independent test. The black lines in the plots represent the fitted curves using second order polynomials. The control case is represented by zero impulse, i.e. no laser treatment. (i) The average growth rate of live cells among attached cells; (j) The average growth rate of live cells among detached cells; the solid line in (i) and (j) corresponds to the normal growth rate of MDA-MB-231, and the dash lines represents the according uncertainty range. The cell growth rate was the ratio of the number of live cells of the second day to that of the first day. The uncertainty of the cell growth was calculated using the standard deviations of the cell doubling time given by Limame et al. [34].

Similar to the WST and cell detachment results, shock waves with very similar peak pressures—the blue dashed boxed case for C/PDMS and the red dashed boxed case for P/PDMS shown in Fig 7(A)—led to very different attached cell death ratios. On the other hand, the cell detachment ratios for both C/PDMS and P/PDMS collapse into the same trend line when plotted against the impulse in Fig 7(E)–7(H). This observation further demonstrated the impulse of shock waves, is the dominant parameter that governs shock impact on cell damage. In addition, the critical impulse threshold of 5 Pa·s seemed to apply to the detached cell death ratio as well. Some detached cells and dead cells were observed in the control group due to exposure to a non-ideal environment during transportation, the shock wave treatment process and cell counting. These non-ideal effects applied to all groups, and the numbers of detached cells and dead cells for control groups were much lower than treatment groups, so these effects did not affect our analysis on the impact of shock waves on cell viability.

Fig 7(A)–7(H) shows that the cell death ratios among the attached cells and detached cells shared similar trends on day 0 and day 1. The cell detachment is correlated to the cell death, but not all detached cells are dead or lost viability, i.e. some detached cells may still have the capability to attach again and proliferate. The values of the cell death ratios generally decreased in day 1 compared to day 0 for both the attached cells and detached cells; the results indicate that the live cells were still capable to divide for both detached and attached cases. A previous study showed that, for MDA-MB-231, the doubling time was 27.8±5.1 hours [34]. Fig 7(I) and 7(J) further demonstrated that the live cells for both the attached and detach cases were able to divide at a rate close to the normal growth rate.

## Discussion

In this work, *in vitro* experimental studies were conducted to show that the impulse of the shock waves, not the peak pressure, was the dominant shock parameter on the viability of MDA-MB-231 cells when the duration of the shock waves was shorter than the fundamental response period of the cells. Shock waves with varying peak pressures, characteristic decay times, and different resulting impulses were generated using two different treatment methods (P/PDMS and C/PDMS coating) on the well plates. The P/PDMS coating yielded higher shock wave impulses, as the subsequent peaks of the P/PDMS coating were stronger than those of the C/PDMS coating, which increased the total interaction duration and total energy imparted into the cells. The reason for the different strengths of subsequent peaks was the energy transmission and reflection in the bottom layer of the well plates. The resultant pressure events permitted an investigation of the relative importance of peak pressure and impulse.

The cell viability was assessed by three assays: WST viability; cell detachment; and cell death of detached and attached cells over two days. Data from all three assays suggested that the shock wave impulse correlated much better with the cell viability than the peak pressure, and hence was deemed the governing shock parameter. This observation was attributed to the fact that the duration of generated shock waves was significantly shorter than the fundamental response period of the MDA-MB-231 cells, as suggested by Rahimzadeh *et al.* [23]. The results also showed that the cell death ratio of detached cells was higher compared to that of attached cells. Moreover, when the shock impulse exceeded 5 Pa·s, the detached cell death ratio on the day of shock wave application exceeded 50%, and this death ratio increased with increasing shock impulse. It should be mentioned that Schmidt *et al.* [19] found a critical pressure threshold of 80 MPa that resulted in 50% survival rate for U87 brain cancer cells. Due to differences of cell lines, certain resultant shock wave characteristics, such as speeds, and cell counting techniques are different between the study by Schmidt *et al.* [19] and the current study, the critical

pressure threshold values might not be comparable. It is possible that when the peak pressure exceeds a very high level, peak pressure can be decisive on cell viability.

This work has demonstrated that the impulse, rather than the peak pressure is the governing shock wave parameter for the viability of MDA-MB-231 cancer cells. The result suggests that a lower-pressure shock wave with a longer duration, or multiple sequential low amplitude shock waves can be applied over a duration shorter than the fundamental response period of the cells to achieve the same impact as shock waves with a high peak pressure but a short duration, as shown in Fig 1. This finding has potential significant implications on how shock waves should be tailored for cancer treatments, enhanced drug delivery, and diagnostic techniques to maximize efficacy while minimizing potential side effects.

Capabilities of the experiment and available instrumentation in this work limited the shock waves generated to a maximum peak pressure of $42.24 \pm 3.89$MPa and shock durations much shorter than the fundamental response period of MDA-MB-231 cells. In addition, we only examined two coating materials for the well plates and a single cell line. Using other coatings will change the acoustic impedance and affect the wave transmission and reflection, which can change the total impulse for a given laser setting. The current study mainly focuses on the live/dead cell counting statistics and comparison of the relative significance of pressure peak and impulse. The mechanisms responsible for the cell death can be complex. Since the approach that we used to generate shock waves can avoid heat and cavitation, the cell viability could be affected by pure mechanical effects. However, the current study did not reveal or conclude the detailed mechanism of how the cell death occurred. Consequently, further studies on cell morphology and changes to intracellular components will be necessary to fully understand the interaction between shock waves and cellular systems. Additional recommendation for future work includes a more systematic investigation to examine shock waves with longer durations, a greater range of peak pressure values, varying time delays between shocks, and testing different cell lines.

## Supporting information

**S1 Table. Pressure data summary.**
(XLSX)

**S2 Table. P/PDMS WST absorbance data summary.**
(XLSX)

**S3 Table. C/PDMS WST absorbance data summary.**
(XLSX)

**S4 Table. Propidium iodide and Hoechst stain data summary.**
(XLSX)

## Acknowledgments

The authors thank Shue Wang, Luciana Rosselli-Murai, Kenneth Ho, Jonathan Estrada, and Andrea Poli for their help with cellular analysis, data analysis, and experiment design. The authors also thank Mathew Boban for his help on the well plate treatment and Professor Anish Tuteja for providing the lab facilities used for the well plate treatment. The authors would also like to acknowledge funding from Mcubed Seed Funding Program provided by the University of Michigan.

## Author Contributions

**Conceptualization:** Yingqian Liao, Ellen M. Arruda, Allen P. Liu, Sofia D. Merajver, Yin Lu Young.

**Data curation:** Yingqian Liao.

**Formal analysis:** Yingqian Liao.

**Funding acquisition:** Ellen M. Arruda, Sofia D. Merajver, Yin Lu Young.

**Investigation:** Yingqian Liao.

**Methodology:** Yingqian Liao, James W. Gose, Allen P. Liu.

**Project administration:** Yin Lu Young.

**Resources:** James W. Gose, Ellen M. Arruda, Allen P. Liu.

**Software:** Yingqian Liao.

**Supervision:** Ellen M. Arruda, Allen P. Liu, Yin Lu Young.

**Visualization:** Yingqian Liao.

**Writing – original draft:** Yingqian Liao.

**Writing – review & editing:** James W. Gose, Ellen M. Arruda, Allen P. Liu, Sofia D. Merajver, Yin Lu Young.

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
