## [Decision Letter · Decision Letter 0]

13 Feb 2020

PONE-D-19-35577

Shock wave impact on the viability of MDA-MB-231 cells

PLOS ONE

Dear Professor Young,

Thank you for submitting your manuscript to PLOS ONE. After careful consideration, we feel that it has merit but does not fully meet PLOS ONE’s publication criteria as it currently stands. Therefore, we invite you to submit a revised version of the manuscript that addresses the points raised during the review process.

We would appreciate receiving your revised manuscript by Mar 29 2020 11:59PM. To enhance the reproducibility of your results, we recommend that if applicable you deposit your laboratory protocols in protocols.io, where a protocol can be assigned its own identifier (DOI) such that it can be cited independently in the future. For instructions see: http://journals.plos.org/plosone/s/submission-guidelines#loc-laboratory-protocols

We look forward to receiving your revised manuscript.

Kind regards,

Corinne Ida Lasmezas

Academic Editor

PLOS ONE

Journal Requirements:

2. Please provide additional information about each of the cell lines used in this work, including source, history, culture conditions and any quality control testing procedures (authentication, characterisation, and mycoplasma testing). For more information, please see http://journals.plos.org/plosone/s/submission-guidelines#loc-cell-lines.

Reviewers' comments:

Reviewer's Responses to Questions

**Comments to the Author**

1. Is the manuscript technically sound, and do the data support the conclusions?

Reviewer #1: Yes

Reviewer #2: Yes

2. Has the statistical analysis been performed appropriately and rigorously? 

Reviewer #1: Yes

Reviewer #2: Yes

3. Have the authors made all data underlying the findings in their manuscript fully available?

Reviewer #1: Yes

Reviewer #2: Yes

4. Is the manuscript presented in an intelligible fashion and written in standard English?

Reviewer #1: Yes

Reviewer #2: Yes

5. Review Comments to the Author

Reviewer #1: The subject of study used by the authors in this manuscript is of great interest. Shock waves, indeed, have been used in many medical applications with great success affecting membrane permeabilization in different cell types and improving drug uptake. The manuscript is presented in a intelligible fashion and written in standard English. The abstract and introduction are both clear and concise, well-referenced and stating that is the impulse, rather than the peak pressure, the shock wave parameter leading to cell detachment, damage and ultimately cell death to the target organisms; data also supported by other authors. Figure 1 clearly shows different shock wave profiles when using the same impulse. Material and methods also explained in detail the different experiments carried out by the authors to demonstrate their hypothesis using two different pretreated well plates with two different coatings (please if C/PDMS coating is supposed to be the control group, this should be clarified in the text). Biological replicates and technical replicates are within a standard range, at least three replicates to assure reproducibility. Moreover, it was important to test whether or not the waves generated to induce cell damage, detachment and cell death were supersonic and therefore, shock waves. Fig. 3(c&d) clearly shows that the pressure applied in the study generated shock waves required to validate their studies. However, figure legends should use a different font compared to the text to make reading easier – this should be considered. Figure 4 and Table 1 show a great difference in peak pressure, decay time and impulse between both coatings that should imply a significant change in cell viability between both P/PDMs and C/PDMs coatings. Standard deviations are small, good statistical analysis with 3 technical replicates and 4 biological replicates per pulse point.

Figure 5 represents the most important experiment of this manuscript since authors show cell viability comparing peak pressure and impulse with both coatings which is actually the goal of this project. WST assay is the right choice to measure cell viability for this specific condition over other assays such as MTT or XTT that have lower sensitivity or are not water-soluble (MTT). Authors claim that “at similar peak pressures, higher normalized absorbance was observed for the C/PDMs coating compared to P/PDMS coating which indicates cell viability”. However, it is important to point out that only for those laser fluence values that imply peak pressures around and over 30MPa, cell viability decreases in P/PDMS compared to C/PDMS. Same comments should be made for the results observed when measuring the impulse. Authors clearly demonstrate that cells plated in P/PDMS coated plate leads to lower cell viability and probably higher cell death compared to C/PDMS coating. However, it took me a while to figure out that the C/PDMS is the control group. If that is the case, it is true that in this figure impulse is the dominant shock parameter on the viability of MDA-MB-231 cells when the shock wave impulse exceeds 5 Pa.s on P/PDMS pre-treated plate. Material and methods as well as introduction should better explain this. Figure 6 and 7 show the same evidence. Finally, discussion is consequent to the data observed in the experiments. If that is not the case, I see no differences between the importance of impulse over peak pressure.

Reviewer #2: The data presentation and interpretation is well done. The results warrant the conclusions.

What is missing in the manuscript are comments with respect to cancer.

Why were MDA-MB-231 one cells (metastatic breast cancer cells) chosen? Which experiments were done with similar methodology on other cancer cells/type? Why is it important to reduce cell viability with Shockwaves? What is the advantage here over biochemical methods I,e, antibodies, drugs who also aim to reduce cell viability?

Did I miss suggestions for mechanisms responsible for cell death through shock waves? They must be given.

For the way forward: is it the aim to apply these shock waves to cancer patients? Has this been done already for other cancer types? If so, are in vivo studies planned with nude mice displaying breast cancer?

The paper needs more citations for the cancer side. I see only 25 references and very little of them citing cancer publications.

Minor: the source for the MBA-MB-231 cells is lacking

6. PLOS authors have the option to publish the peer review history of their article (what does this mean?). If published, this will include your full peer review and any attached files.

Reviewer #1: Yes: Juan Carlos Jado

Reviewer #2: No

---

## [Author Response · Author response to Decision Letter 0]

30 Mar 2020

Response to reviewer #1:

The authors would like to thank the reviewer for the comments and the suggestions for improvement. 

Reviewer #1: The subject of study used by the authors in this manuscript is of great interest. Shock waves, indeed, have been used in many medical applications with great success affecting membrane permeabilization in different cell types and improving drug uptake. The manuscript is presented in a intelligible fashion and written in standard English. The abstract and introduction are both clear and concise, well-referenced and stating that is the impulse, rather than the peak pressure, the shock wave parameter leading to cell detachment, damage and ultimately cell death to the target organisms; data also supported by other authors. Figure 1 clearly shows different shock wave profiles when using the same impulse. Material and methods also explained in detail the different experiments carried out by the authors to demonstrate their hypothesis using two different pretreated well plates with two different coatings (please if C/PDMS coating is supposed to be the control group, this should be clarified in the text). Biological replicates and technical replicates are within a standard range, at least three replicates to assure reproducibility. Moreover, it was important to test whether or not the waves generated to induce cell damage, detachment and cell death were supersonic and therefore, shock waves. Fig. 3(c&d) clearly shows that the pressure applied in the study generated shock waves required to validate their studies. 

Response: Thank you for the positive review. The untreated cells corresponded to the control group. We explained that in the WST cell viability test subsection in the Material and methods section. We have added clarifications in other parts in the revised manuscript, including the WST absorbance results subsection and the associated figure caption.

However, figure legends should use a different font compared to the text to make reading easier – this should be considered. 

Response: Thank you for the suggestions. We have changed the legends to italic font and reduced the line spacing to make it easier to read. 

Figure 4 and Table 1 show a great difference in peak pressure, decay time and impulse between both coatings that should imply a significant change in cell viability between both P/PDMs and C/PDMs coatings. Standard deviations are small, good statistical analysis with 3 technical replicates and 4 biological replicates per pulse point.

Figure 5 represents the most important experiment of this manuscript since authors show cell viability comparing peak pressure and impulse with both coatings which is actually the goal of this project. WST assay is the right choice to measure cell viability for this specific condition over other assays such as MTT or XTT that have lower sensitivity or are not water-soluble (MTT). Authors claim that “at similar peak pressures, higher normalized absorbance was observed for the C/PDMs coating compared to P/PDMS coating which indicates cell viability”. However, it is important to point out that only for those laser fluence values that imply peak pressures around and over 30MPa, cell viability decreases in P/PDMS compared to C/PDMS. Same comments should be made for the results observed when measuring the impulse. 

Response: We believe the reviewer is referring to the results shown in Fig. 5(a) with respect to the comment about 30 MPa. Fig. 5(a) shows that for peak pressures greater than 30 MPa, the trends deviate for the P/PDMS and C/PDMS coatings, where the results for the C/PDMS coating appears to plateau, while the normalized WST absorbance decreases as the pressure increases beyond 30 MPa. Similar differences between the P/PDMS and C/PDMS coatings are observed in Fig. 7. The reason for the deviation in trends between the P/PDMS and C/PDMS when the peak pressure exceeds 30 MPa is because of the different laser fluence value when the critical impulse threshold of 5 Pa·s is reached. As shown in Fig. 4, for the P/PDMS coating, the 5 Pa·s impulse threshold is reached when the laser fluence exceeds ~0.28 J/cm2, which corresponds to a peak pressure ~30 MPa. On the other hand, for the C/PDMS coating, the critical impulse threshold of 5 Pa·s is projected to be reached when the laser fluence reaches higher than ~0.6 J/cm2, which corresponds to a peak pressure higher than ~35 MPa for the C/PDMS coating. Hence, when the peak pressure exceeds ~ 30 MPa, the trends deviate for the cell damage for the C/PDMS and P/PDMS coatings when plotted against the peak pressure. As shown in Figs. 5 and 7, the trends for both coatings are more consistent when plotted against the impulse. A clarification has been added to the revised manuscript.

Authors clearly demonstrate that cells plated in P/PDMS coated plate leads to lower cell viability and probably higher cell death compared to C/PDMS coating. However, it took me a while to figure out that the C/PDMS is the control group. If that is the case, it is true that in this figure impulse is the dominant shock parameter on the viability of MDA-MB-231 cells when the shock wave impulse exceeds 5 Pa.s on P/PDMS pre-treated plate. Material and methods as well as introduction should better explain this. Figure 6 and 7 show the same evidence. Finally, discussion is consequent to the data observed in the experiments. If that is not the case, I see no differences between the importance of impulse over peak pressure.

Response: We had untreated cells (without shock waves applications) as the control group, but the reviewer is correct that we used different coatings, P/PDMS and C/PDMS. As shown in Fig. 4, as the laser fluence increases, the difference in shock impulse values between the P/PDMS and C/PDMS coating are much greater than the difference in peak pressure values. For the P/PDMS coating, when the peak pressure reaches 30MPa, the impulse reaches the critical threshold of 5 Pa.s; while for the C/PDMS coating, the impulse is much lower with a similar peak pressure. The results in Figs. 5-7 show that the WST absorbance and cell death correlate much better with impulse than the peak pressure. Hence, we conclude that the shock impulse is better correlated to cell viability than the peak pressure. We have made changes in the revised manuscript to make it clearer that we had untreated cells as the control group in the WST tests.

Response to reviewer #2:

The authors would like to thank the reviewer for the comments and the suggestions for improvement. 

Reviewer #2: The data presentation and interpretation is well done. The results warrant the conclusions.

What is missing in the manuscript are comments with respect to cancer.

Response: Thank you for the positive feedback and suggestion. We have made changes in Introduction and Discussion in the revised manuscript to explicitly comment on cancer.

Why were MDA-MB-231 one cells (metastatic breast cancer cells) chosen? 

Response: MDA-MB-231 is a commonly used cell line in cell biology research, and one that was available and previously studied in our lab [1]. 

Which experiments were done with similar methodology on other cancer cells/type? 

Response: In the Discussion section, we noted that a study with similar methodology was done by Schimidt et al. [2] with U87 brain cancer cells.

Why is it important to reduce cell viability with Shockwaves? 

Response: Shock waves can be applied non-intrusively with high precision via extracorporeal techniques, which is an important potential advantage over invasive surgery [3,4]. We have added a statement in the Introduction to clarify this point.

What is the advantage here over biochemical methods I,e, antibodies, drugs who also aim to reduce cell viability?

Response: Shock wave techniques has the potential advantages of being a non-invasive, targeted, extracorporeal cancer treatment method [5]. However, the objective of this work is not to compare the pros and cons of the various methods. Rather, the objective is to investigate the relative significance of different mechanical characteristics of shock waves on cell viability, as shock waves have been used to treat cancer, kidney and gall stones[6], as well as for enhanced drug delivery [7–9] and a variety of diagnostic techniques [3,4]. This study is important, as current treatment procedures typically use shock waves with a very high peak pressure, which can have negative side effects. The results show that the shock impulse, instead of the peak pressure, governs the shock impact on cell viability, which suggests that multiple low amplitude shock waves can be applied over a short duration to minimize the side effects while still ensuring efficacy. We have revised the manuscript to clarify these points.

Did I miss suggestions for mechanisms responsible for cell death through shock waves? They must be given.

Response: The objective of this work is to study the impact of mechanical parameters on cell viability, but not to determine the specific mechanisms that cause cell death. In the discussion, we had recommendations to conduct additional research to study the mechanisms responsible for cell death.

For the way forward: is it the aim to apply these shock waves to cancer patients? Has this been done already for other cancer types? If so, are in vivo studies planned with nude mice displaying breast cancer?

Response: While the authors would like to extend the work for in vivo studies, we do recommend additional in vitro studies to first identify the cell damage mechanisms, and to repeat these studies for multiple cell lines and for a wider range of shock wave peak pressure and impulse values prior to in vivo studies in mice. These recommendations have been added to the end of the paper.

The paper needs more citations for the cancer side. I see only 25 references and very little of them citing cancer publications.

Response: We have added citations related to cancer in the revised manuscript.

Minor: the source for the MBA-MB-231 cells is lacking

Response: We have added the source for the MBA-MB-231 cells in subsection Cell culture.

References:

1. Rosselli-Murai LK, Yates JA, Yoshida S, Bourg J, Ho KKY, White M, et al. Loss of PTEN promotes formation of signaling-capable clathrin-coated pits. J Cell Sci. 2018. doi:10.1242/jcs.208926

2. Schmidt M, Kahlert U, Wessolleck J, Maciaczyk D, Merkt B, Maciaczyk J, et al. Characterization of a setup to test the impact of high-amplitude pressure waves on living cells. Sci Rep. 2014;4: 1–9. doi:10.1038/srep03849

3. Maloney E, Hwang JH. Emerging HIFU applications in cancer therapy. International Journal of Hyperthermia. 2015. doi:10.3109/02656736.2014.969789

4. Steinhauser MO. On the Destruction of Cancer Cells Using Laser-Induced Shock-Waves: A Review on Experiments and Multiscale Computer Simulations. Radiol - Open J. 2016;1: 60–75. doi:10.17140/ROJ-1-110

5. Chen H, Zhou X, Gao Y, Zheng B, Tang F, Huang J. Recent progress in development of new sonosensitizers for sonodynamic cancer therapy. Drug Discovery Today. 2014. doi:10.1016/j.drudis.2014.01.010

6. Bailey MR, Khokhlova VA, Sapozhnikov OA, Kargl SG, Crum LA. Physical mechanisms of the therapeutic effect of ultrasound (a review). Acoustical Physics. 2003. doi:10.1134/1.1591291

7. Etame AB, Diaz RJ, Smith CA, Mainprize TG, Hynynen K, Rutka JT. Focused ultrasound disruption of the blood-brain barrier: a new frontier for therapeutic delivery in molecular neurooncology. Neurosurg Focus. 2012;32: E3. doi:10.3171/2011.10.FOCUS11252

8. Jagadeesh G, Prakash GD, Rakesh SG, Allam US, Krishna MG, Eswarappa SM, et al. Needleless vaccine delivery using micro-shock waves. Clin Vaccine Immunol. 2011. doi:10.1128/CVI.00494-10

9. Chen PY, Hsieh HY, Huang CY, Lin CY, Wei KC, Liu HL. Focused ultrasound-induced blood-brain barrier opening to enhance interleukin-12 delivery for brain tumor immunotherapy: A preclinical feasibility study. J Transl Med. 2015;13: 1–12. doi:10.1186/s12967-015-0451-y

---

## [Decision Letter · Decision Letter 1]

20 May 2020

Shock wave impact on the viability of MDA-MB-231 cells

PONE-D-19-35577R1

Dear Dr. Young,

We are pleased to inform you that your manuscript has been judged scientifically suitable for publication and will be formally accepted for publication once it complies with all outstanding technical requirements.

With kind regards,

Corinne Ida Lasmezas

Academic Editor

PLOS ONE

Additional Editor Comments (optional):

Reviewers' comments:

Reviewer's Responses to Questions

**Comments to the Author**

1. If the authors have adequately addressed your comments raised in a previous round of review and you feel that this manuscript is now acceptable for publication, you may indicate that here to bypass the “Comments to the Author” section, enter your conflict of interest statement in the “Confidential to Editor” section, and submit your "Accept" recommendation.

Reviewer #1: All comments have been addressed

Reviewer #2: All comments have been addressed

2. Is the manuscript technically sound, and do the data support the conclusions?

Reviewer #1: Yes

Reviewer #2: Yes

3. Has the statistical analysis been performed appropriately and rigorously? 

Reviewer #1: Yes

Reviewer #2: Yes

4. Have the authors made all data underlying the findings in their manuscript fully available?

Reviewer #1: Yes

Reviewer #2: Yes

5. Is the manuscript presented in an intelligible fashion and written in standard English?

Reviewer #1: Yes

Reviewer #2: Yes

6. Review Comments to the Author

Reviewer #1: The authors have adequately answered all comments about the manuscript. Text has been modified as expected and been made clearer.

Reviewer #2: May queries have been fully addressed. I recommend publication as it stands. I have no concerns about dual publication, research ethics or publication ethics.

7. PLOS authors have the option to publish the peer review history of their article (what does this mean?). If published, this will include your full peer review and any attached files.

Reviewer #1: Yes: Juan Carlos Jado

Reviewer #2: Yes: Prof SFT Weiss

---

## [Editor Report · Acceptance letter]

26 May 2020

PONE-D-19-35577R1 

Shock wave impact on the viability of MDA-MB-231 cells 

Dear Dr. Young:

I am pleased to inform you that your manuscript has been deemed suitable for publication in PLOS ONE. Congratulations! Your manuscript is now with our production department. 

With kind regards,

on behalf of

Dr. Corinne Ida Lasmezas 

Academic Editor

PLOS ONE